# Effect of Iodine Filler on Photoisomerization Kinetics of Photo-Switchable Thin Films Based on PEO-BDK-MR

**DOI:** 10.3390/polym13050841

**Published:** 2021-03-09

**Authors:** Qais M. Al-Bataineh, A. A. Ahmad, A. M. Alsaad, I. A. Qattan, Ihsan A. Aljarrah, Ahmad D. Telfah

**Affiliations:** 1Department of Physics, Jordan University of Science & Technology, P.O. Box 3030, Irbid 22110, Jordan; qalbataineh@ymail.com (Q.M.A.-B.); sema_just@yahoo.com (A.A.A.); ihsanaljarrah@gmail.com (I.A.A.); 2Department of Physics, Khalifa University of Science and Technology, Abu Dhabi P.O. Box 127788, United Arab Emirates; issam.qattan@ku.ac.ae; 3Leibniz Institut für Analytische Wissenschaften-ISAS-e.V., Bunsen-Kirchhoff-Straße 11, 44139 Dortmund, Germany; telfah.ahmad@isas.de; 4Hamdi Mango Center for Scientific Research (HMCSR), The Jordan University, Amman 11942, Jordan

**Keywords:** Polyethylene Oxide (PEO), Benzyl Dimethyl Ketal (BDK), Methyl-Red (MR), Iodine (I_2_), complex composite thin films, photoisomerization processes, molecular solar thermal energy storage media

## Abstract

We report the effect of an iodine filler on photoisomerization kinetics of photo-switchable PEO-BDK-MR thin films. The kinetics of photoisomerization and time progression of PEO-BDK-MR/I_2_ nanocomposite thin films are investigated using UV-Vis, FTIR spectroscopies, and modified mathematical models developed using new analytical methods. Incorporating iodine filler into the PEO-BDK-MR polymeric matrix enhances the isomerization energy barrier and considerably increases the processing time. Our outcomes propose that enhanced photoisomerized and time processed (PEO-BDK-MR)/I_2_ thin films could be potential candidates for a variety of applications involving molecular solar thermal energy storage media.

## 1. Introduction

Photo-switchable devices offer a new insight into several optical applications such as photonic transistors, solar storage systems, and cancer therapy [1,2,3,4]. There are many kinds of photo-switchable materials, including azo-benzenes [5], tetracarbonylfulvalenediruthenium complexes [6], norbornadienes (NBDs) [7], dihydroazulenes [8], and fulvalene dirutheniums [9]. Azobenzenes are the most preferred molecule due to their simple chromophore structure (sensitive to light) [10], high quantum yields, and can be switched through a reversible (trans-cis) isomerization using a low intensity of light irradiation [11,12,13]. Moreover, the efficiency of the azo molecule has four main challenges represented by incomplete photoswitching due to overlapping absorbance in the irradiation regions of interest [14], short storage lifetime due to the instability of the metastable state (cis state), low-energy-density afforded by isomerization enthalpy, and low-temperature level of the released heat associated with low power density [1]. Siewertsen et al. [15] recently identified such a system, i.e., an azobenzene covalently bridged in ortho-positions by an ethylene linker, in which a 100 nm splitting of the n → π* transitions of *trans*- and *cis*- isomers allows for switching in both directions with visible light and almost complete photoconversions (100% *trans* → *cis* and 92% *cis* → *trans*). Moreover, Hecht et al. [16] recently reported that 2,2,6,6-tetrafluoroazobenzene shows near quantitative photoswitching and the longest thermal half-life reported for an azobenzene molecule (∼700 days at 25 °C in DMSO). Z. Zhang et al. [11] introduced pyrazolylazophenyl ethers (pzAzo ethers) as a class of azo photoswitches that provides quantitative (>98%) trans–cis photoisomerization (365 nm light), near-quantitative (95–96%) reverse isomerization (532 nm light), and a long cis-isomer half-life of three months.

In our previous works [17,18,19,20], we presented a new insight into the kinetics of the photoisomerization process and the evolution of hybrid thin films considering the azo-dye methyl red (MR) as a photo-switchable material by studying the absorbance spectra as well as FTIR spectroscopy. This work’s main theme is to investigate the effect of iodine filler (I_2_) on photoisomerization kinetics of photo-switchable thin films based on PEO-BDK-MR. Iodine was used to synthesise polymer–halogen complexes, which affects their optical, electrical, and dielectric properties [21]. The iodine dopant ions in any polymer may be present at many sites, such as exchange into the polymer chains, or reside at the amorphous/crystalline boundaries, forming charge transfer compounds. Moreover, it may form ions aggregates between the polymer chains [22], which may increase the lifetime, activation energy and, consequently, increase thermal storage energy according to increasing photochemical potential.

## 2. Theoretical Background

The photoisomerization mechanisms of the *trans-cis* isomer and thermally induced reverse *cis-trans* isomer in complex composites play a significant role in interpreting the physical, electric, structural, and optical properties of azo-benzenes [23,24,25]. The photoisomerization process of complex composites is crucial in obtaining a more in-depth insight into photochromic molecular switches’ functioning, light-gated transistors, molecular solar thermal energy storage media, optical data storage devices, and heat and light sensors. The separation between the two carbon atoms in aromatic rings of methyl red (MR) is contracted from 9.0 Å in the *trans*-phase to 5.5 Å in the *cis*-phase as a result of the photoisomerization process, as illustrated in Figure 1 [26]. The azo-benzenes that utilize *trans*-phase are fundamentally flat and show no dipole moment, while the azo-benzenes exhibit *cis*-phase assumes an angular geometry and a notable dipole moment of 3.0 Debye [27].

The proficient and revocable photoisomerization between a thermally stable *trans*-state and a metastable *cis*-state under the illumination and thermal relaxation is of prime importance for fundamental and practical aspects [28,29]. The *trans*-phase of the isomers is thermally stable at room temperature [30,31]. The azo-benzenes substituents mainly determine the major transitions that occur within the azo-benzene [32]. Usually, the isomer undergoes a series of symmetric or antisymmetric non-breakable bending of one of the aromatic rings. Thus, twisting leads to prolongation and compression of the mono-molecule of the isomer, finally generating successive variations in the azo-benzenes’ absorption band [33,34].

Azobenzene undergoes *trans*-*cis* isomerization by $S_{1}\leftarrow S_{0}$ and $S_{2}\leftarrow S_{0}$ excitations and *cis-trans* isomerization by exciting into the $S_{1}$ or $S_{2}$ states [35]. Isomerization in stilbene is associated mainly due to rotation, with quantum yield equaling unity [36]. The four-level model representing the *trans*- and the *cis*- photoisomerization has been entirely studied by Al-Bataineh Q.M. et al. [17,18]. As the AZO molecule in its *trans*-form is irradiated, it transforms to the *trans* metastable excited state. Shortly later, the *trans*-isomers transform into the *cis*-state. A quantum yield of $\varphi_{TC}$ characterizes the conversion. Moreover, the unexcited *cis*-isomers absorb light to transmute to the excited *cis*-state. Finally, the *cis*-isomers return to the *trans*-state either by *cis-trans* photoisomerization with a quantum yield $\varphi_{CT}$ or by thermal relaxation characterized by a rate *k*. The whole process is entirely described by $\sigma_{T}$ and $\sigma_{C}$ absorption cross-sections for the *trans*- and *cis*- forms, respectively, as illustrated in Figure 2.

To obtain a deeper insight into the kinetics of photoisomerization of the AZO molecule, photoisomerization rate and thermal/optical activation barrier for *trans-cis* photoisomerization is evaluated [37,38]. The photoionization rate $\left(p\right)$ can be given as:(1)$$\ln\frac{A_{t}-A_{\infty }}{A_{0}-A_{\infty }}=-pt$$
where $A_{0},\;A_{t}$, and $A_{\infty }$ are the absorbance preceding to the light exposure, during radiation, and after the light exposure for a long time, respectively. Furthermore, the variations in p are related to the disparity of $\Delta E_{a}$ [39] as:(2)$$\Delta E_{a}=-kT\ln\frac{h\ln2}{\tau_{1/2}kT}$$
where $\tau_{1/2}$ is the time required to transfer half the *trans*-states into *cis*-states; $\tau_{1/2}=\ln2/p$, k is Boltzmann constant, h is Plank constant, and T is the temperature. Therefore, Equation (2) becomes:(3)$$\Delta E_{a}=-kT\ln\frac{hp}{kT}$$

The enhanced absorption of energetic solar power is envisaged by strong electronic transitions in the visible region. The amount of energetic-absorbed solar energy doubles or triples the activation barrier-energy $\Delta E_{i}$. The absorption of such energy ensures a total isomerization of AZO-molecules [40]. The energy (ΔH) for *cis*-isomerization of AZO-molecules must be <$\Delta E_{i}$. Indeed, $\Delta H=\Delta E_{i}-\Delta E_{a}$ where, $\Delta E_{i}$ is the activation barrier energy required to isomerize one AZO molecule and $\Delta E_{a}$ is associated with thermal and optical activation barrier and the storing efficiency that is typically less than 30% [39,40].

The amount of energy stored in a thermally induced solar system usually depends on the difference between two states’ energies, mainly the *cis* and *trans* states. For thermally isomerized AZO molecule, *trans*-isomer exhibits lower energy than the *cis*-isomer (Figure 3) [41]. Ideally, the thermal reversal process barrier has to be of enough height to warrant long storage times. *Trans*-azobenzene has to absorb many electromagnetic waves from the sun and block all of the light in the *cis*-azobenzene. Alternatively, the quantum yield for the transformation from *trans*-to *cis*-azobenzene should be very high [42]. Unfortunately, *trans*-azobenzene absorbs UV-light. However, the storing process should occur in the visible region by adjusting the exchange form [43]. Various attempts have been introduced to enhance the *trans*-*cis* energy difference and thus increase the stored energy. Among these criteria is modification of stabilizing groups by subtle exchanges such as London dispersion [44], integration of the AZO molecule into macrocyclic structures to exploit strain energy in the *cis*-isomer [45,46], or AZO can be linked to carbon nanotubes to build up storage energy [47].

## 3. Experimental Details

### 3.1. Materials

300.000 g/mol Polyethylene Oxide (PEO) (–CH_2_CH_2_O–)_n_
256.30 g/mol Benzyl Dimethyl Ketal (BDK) 126.90 g/mol Iodine (I_2_)32.04 g/mol Absolute Methanol (CH_3_OH, with a purity of 99.8%)The above materials were purchased from Sigma-Aldrich Co., Inc., Munich, Germany.SCP SCIENCE (Montreal, QC, Canada) supplied Methyl-Red (MR) (C_15_H_15_N_3_O_3_) of pH level between 4.2 and 4.6 powder.

### 3.2. Synthesis of (PEO-BDK-MR)/I_2_ Complex Composite Thin Films

The solvent used to prepare the solutions is absolute methanol. We dissolve 1 g of PEO in 100 mL methanol at 45 °C and place the resulting solution on a magnetic stirrer for five hours. In addition, 1 g of BDK and 1 g of MR are sequentially dissolved in 100 mL methanol at room temperature. The resulting solutions are subjected to a continuous magnetic stirring for 2 h. The PEO-BDK-MR polymer composite solution is obtained by mixing PEO solution, BDK solution, and MR solution in a 5:5:1 volumetric ratio using a magnetic stirrer for 6 h. Then, all complex composites undergo sonication for 6 h to ensure homogeneity. The (PEO-BDK-MR)/I_2_ complex composite solution is prepared by incorporating I_2_ powder into PEO-BDK-MR polymeric matrix for three different I_2_ concentrations (0.1%, 1%, and 10%). Finally, (PEO-BDK-MR)/I_2_ complex composite thin films are prepared by casting technique on glass substrates. The solvents are dispersed, and the organic residues are eliminated by air drying at room temperature for 24 h.

### 3.3. Characterization

A Double-Beam UV–vis Spectrophotometer (U-3900H) is used to measure the absorbance of as-prepared thin films at room temperature. The vibrational bands are identified using the Fourier transform infrared spectroscopy (FTIR) (Bruker Tensor 27 spectrometer with a disc of KBr). The (PEO-BDK-MR)/I_2_ complex composite thin films are irradiated by UV-light source (λ= 366 nm wavelength, power = 6 Watts (32 W/cm^2^ of intensity)) for 0, 1, 2, …, 10 min to investigate the influence of the time-evolution of the transformation from *trans*-phase to *cis*-phase. The absorbance spectra are measured each time the films are exposed to UV-irradiation. The films are then subjected to a thermal relaxation process under atmospheric conditions at 90 °C for 60 min to complete the cycles and study the transformation from *cis*-phase to *trans*-phase. Furthermore, FTIR spectroscopy is used to examine the UV-irradiation’ influence by identifying the vibrational changes in the *trans*- and cis-phases’ bonding modes.

## 4. Results and Discussion

### 4.1. Photoisomerization Kinetics of PEO-BDK-MR

The kinetics of the photoisomerization process of the PEO-BDK-MR polymer composite thin film is investigated by exploring the transformation from the initial *trans*-phase to *cis*-phase via UV-illumination. Figure 4a shows the absorbance spectra of PEO-BDK-MR polymer composite thin films illuminated by UV-light for different exposure periods. The major absorption peak at the initial *trans*-phase in the visible range was at 419 nm with an absorbance amplitude of 0.619. The dominant absorbance peak located in the spectral range 370–600 nm is associated with the $\pi –\pi^{*}$ electronic transition band. The film was exposed to UV-light for 1, 2, 3, 4, and 5 min to obtain a more in-depth insight into the phase transformation’s time evolution. Moreover, the absorbance peak amplitude in the visible region (370–600 nm) has significantly reduced by blue shifting, transferring the *trans*-phase into a *cis*-phase as predicted. The photoisomerization process is a multi-stage process [17]. The absorbance spectra examined before and after UV-exposure confirms the presence of double absorbance bands that occurred close to 415 and 475 nm, respectively. A prolonged illumination does not result in any significant change in the absorption spectra, emphasizing that a quasi-state of the photo-stationary phase occurs between the *trans*- and the *cis*-states. Electronic transitions in the visible spectrum enhance the performance of the film in high-energy solar-power applications.

Irradiation of the material in the *trans*-isomerization state by UV-light stimulates the conversion to the *cis*-isomerization state. The *cis*-isomerization state undergoes a reversible conversion to the *trans*-isomerization state thermally or/and optically. The PEO-BDK-MR composite thin films are irradiated with optimum UV-light to promote the trans-state transformation to the *cis-state*. The thermal relaxation process effectively initiates the reversible transformation from cis-state to trans-state. The cycle can be periodically repeated several times. The inset of Figure 4a shows a periodically repeated photoisomerization confirming reliable hysteresis cycles for PEO-BDK-MR composite thin films with a slight distinctive-loss. The first-order kinetics of photoisomerization can be further understood by computing the photoisomerization rate and the thermal/optical activation barrier $\Delta E_{a}$
*trans-cis* photoisomerization Equation (1) [37,38]. The average value of $\ln\left(A_{t}-A_{\infty }/A_{0}-A_{\infty }\right)$ of the PEO-BDK-MR polymer composite thin film is plotted versus the time (*t*), as shown in Figure 4b. The photoisomerization rate constant (*p*) is obtained from the slope to be $p=4.03\;\times \;10^{-3}\;s^{-1}$. Furthermore, the changes in p could be plausibly attributed to the variation of $\Delta E_{a}$ [39] and determined using Equation (2). The half-life is found to be $\tau_{1/2}=2.89\;\min$ for films and consequently, $\Delta E_{a}=2.011\;\mathrm{eV}$. It has been reported that *trans*-*cis* photoisomerization is attributed to the steric effect, and hence, the reaction barriers are predominant by the electronic arrangement of the −N=N− group [48]. 

Figure 4a suggests that each absorbance peak of the PEO-BDK-MR polymer composite contains two absorption peaks (two frequency bands). As the UV-illumination time increases, the absorbance amplitude decreases, confirming that trans-phase transmutation to *cis*-phase has occurred appropriately. Increasing the UV-light exposure time boosts newly configured dual-shapes in the absorption peaks, as shown in Figure 4a, implying the development of hindered dual-frequency bands forming two prominent crests instead of one rough absorption peak. Further confirmation of the occurrence of two-fold overlapped sub-peaks is shown from the superficial shoulders created mainly at the lower and higher frequency spectrum portions at 415 nm and 475 nm. Figure 4c–h shows the well-known PEO-BDK-MR thin-film peaks for all UV-exposure durations in the visible range fitted to pair Gaussian crests [18,49]. Figure 4c represents the PEO-BDK-MR prominent peak at the initial *trans*-phase that contributed to the coupled Gaussian crests. The absorption spectra exhibit two bands (labeled as high and low-frequency) with peaks at around 416 and 476 nm and linewidths of 48.52 and 108.12 nm. The absorbance spectrum expands in its shaped form as the UV-illumination exposure time varies, as shown in Figure 4c–h, inducing the transmutation from *trans*-state to *cis*-state.

A detailed numerical inspection of the lower- and higher-frequency bands was executed to achieve more profound insight into the significance of UV-illumination time. The amplitude and area variation beneath the absorbance trajectory are considered in Figure 4i,j during the transmutation from *trans*-phase into *cis*-phase. The alteration of the sub-bands is closely related to the UV-exposure performed within the methodical discrepancy from the linewidth in a proper Gaussian function. The area and amplitude of the overall absorbance and the low-frequency bands regularly drop with time. On the contrary, the high-frequency band was halted during the first minute of illumination and then allotted a slight decrease. Furthermore, the low-frequency sub-band has undergone a blue-shifting suggesting a bathochromic variation caused by light absorbed by MR, which is regarded as an H-bond donor origin or through dissociated-intermediate H-bond donors root [50,51]. Consequently, the extended absorption band occurs due to H-bond’s mutual interaction amongst the PEO and the azo-nitrogen contents mediated by MR molecules. The blue-shifting for both sub-bands (low- and high-frequency bands) demonstrates similar tendencies, indicating that H-bonds directly influence different sites of MR molecules. Additionally, the high-frequency band exhibits steady bathochromic shifts during the UV-illumination, providing two distinct configurations at the H-bonds’ constructions depicted by MR transmutation. The shift in bands’ amplitude guarantees that the time-dependent photoisomerization process is a multi-step [52]. The process of photoisomerization is a multifaceted response to the four-level *(trans*- and *cis*- isomerization) representation as reported by Sekkat Z. et al. [29] and Lee G.T. et al. [53] (see Figure 2).

Figure 5 shows the FTIR spectra of PEO-BDK-MR polymer composite thin film for the *trans*- and *cis*-states in the spectral range of 400–4000 cm^−1^. The spectral peak for the bending C–H bond occurs in the spectral range 400–1000 cm^−1^, bending O–CH_3_ recognized in the 1000–1100 cm^−1^ range, stretching C–O bond is observed in the 1100–1300 cm^−1^ range, C–N bond is found at 1370 cm^−1^, bending –CH_2_ bond located at 1470 cm^−1^, C=O bond positioned at 1600 cm^−1^, N=N bond situated in the1690–1740 cm^−1^ range and stretching C–H placed at 2900 cm^−1^ are observed. Furthermore, the absorbance spectra for *cis*- and *trans*-phases of PEO-BDK-MR exhibit similar preferences, with tiny parts of the bands slipped to lower energies. This reflection verifies that such interactions significantly influence N=N bands at frequencies in the spectral 1650–1750 cm^−1^ range. The obtained intensity in the infrared spectrum of the N=N band for the *cis*-MR is enormously higher in comparison with that for the *trans*-MR, affirming the enhanced photoisomerization rule of the PEO-BDK-MR polymer composite thin film.

### 4.2. Photoisomerization Kinetics of (PEO-BDK-MR)/I_2_ (0.1%)

Figure 6a shows the absorbance spectra of (PEO-BDK-MR)/I_2_ (0.1%) complex composite thin film illuminated by UV-light for different durations of time. The dominant absorption peak at the initial *trans*-phase in the visible range was at 425 nm with an absorbance amplitude of 0.393. The main absorbance peak located in the 360–600 nm range is referred to the $\;\pi –\pi^{*}$ electron transition band. The film is then exposed to UV-light for 1, 2, 3, 4, 5, 6, and 7 min to reveal the time evolution of the investigated film’s photoisomerization. The absorbance spectra of the PEO-BDK-MR/I_2_ with complex composite thin film with a 1% ratio affirms that double absorbance bands appeared near 416 and 474 nm before and after the UV-exposure. For time exposure of more than 5 min, a new peak appeared at 528 nm associated with the formation of I_2_ in the polymer matrix. 

The inset in Figure 6a displays the hysteresis cycles for (PEO-BDK-MR)/I_2_ (0.1%) thin films. Figure 6b shows $\ln\left(A_{t}-A_{\infty }/A_{0}-A_{\infty }\right)$ as a function of exposure time. The obtained value of photoisomerization rate is $p=3.40\;\times \;10^{-3}\;s^{-1}$. The half-life, $\tau_{1/2}=3.39\;\min$, and $\Delta E_{a}=2.015\;\mathrm{eV}$. Figure 6a suggests that each absorbance peak consists of two absorption peaks (two frequency bands). It indicates a designated dual-shape in the absorption peaks as films are exposed to UV-light for a longer time. Two-fold coincided sub-peaks appear at 416 nm and 474 nm with linewidths of 46.62 and 108.76 nm. As demonstrated in Figure 6c–h, two-fold absorbance peaks of (PEO-BDK-MR)/I_2_ (0.1%) exposed to UV light in the visible range could be fitted to pair Gaussian crests [18]. In particular, Figure 6c represents the projecting peak at the initial *trans*-phase that subsidizes the major contribution to the coupled Gaussian crests.

Figure 6i,j show the variation of the amplitude of absorbance peak in the transformation from *trans*-phase into *cis*-phase. The area and the amplitude of the absorbance peak and high-frequency bands decay uninterruptedly with time. However, the area under the high-frequency band increases as a function of exposure time owing to the formation of $I_{3}^{-}$ inside the polymeric matrix. The blue-shift experienced by the low-frequency sub-band could be attributed to the dissociation of the transitional H-bond donor’s root [50,51]. The blue-shift for the low- and high-frequency confirms H-bonds’ effect on locating the exact site of MR molecules. The multi-step time-dependent photoisomerization process is confirmed by the bands’ amplitude shifts [52].

Figure 7 shows the FTIR spectra for the *trans*- and *cis*-states of (PEO-BDK-MR)/I_2_ (0.1%) in the spectral range of 400–4000 cm^−1^. The C–H and C–I bond bending occurs in the spectral range of 400–1000 cm^−1^. Furthermore, the absorbance spectra of the *cis*- and *trans*-phases have parts of the bands shifted towards the lower energy-regions, indicating that such band-slipping significantly influences the N=N bands in the 1650–1750 cm^−1^ range. The obtained intensity in the infrared spectrum of the N=N band for the *cis*-MR is exceptionally high in comparison with that of the *trans*-MR, verifying the enhancement of photoisomerization of the (PEO-BDK-MR)/I_2_ (0.1%) complex composite thin film.

### 4.3. Photoisomerization Kinetics of (PEO-BDK-MR)/I_2_ (1.0%)

Figure 8a shows the absorbance spectra of UV-illuminated (PEO-BDK-MR)/I_2_ (1.0%) complex composite thin films exposed continuously for exposure periods of 1, 2, 3, 4, 5, 6, 7, 8, 9, and 10 min. The major absorption peak at the initial *trans*-phase was at 423 nm with an amplitude of 0.418. The absorbance peak in the spectral 360–600 nm range is associated mainly with the $\;\pi –\pi^{*}$ electronic transition band. The photoisomerization process is steady and takes place via several sub-stages [17]. The double-band absorbance spectra near 416 and 478 nm appeared upon illuminating the films with UV light for short exposure periods. However, upon exposing the films to the UV light for longer periods (>5 min), an absorption peak appears at 528 nm associated with the formation of I_2_ in the polymer matrix. The inset in Figure 8a shows a regularly repeated photoisomerization confirming steadfast hysteresis cycles with negligible dissipation. Utilizing Figure 8b, the photoisomerization rate constant $\left(p\right)$ is found to be $p=1.70\;\times \;10^{-3}\;s^{-1}$ and $\tau_{1/2}=6.80\;$ min, and thus, $\Delta E_{a}=2.033\;\mathrm{eV}$. As anticipated, p  and $\tau_{1/2}$ exhibit values twice and halved of that of (PEO-BDK-MR)/I_2_ (0.1%) thin films. The value of $\Delta E_{a}$ indicates that reaction barriers are controlled by the −N− group [48]. 

Figure 8a demonstrates that the absorbance peaks exhibit two-fold frequency. As the UV-exposure time increases, the absorbance amplitude decreases, suggesting that trans-phase transmutation to *cis*-phase occurs accordingly. Increasing the exposure time induces a constructed dual-shape in the absorption peaks leading to the development of an impeded dual-frequency bands forming two distinct crests instead of one rough absorption peak. Further confirmation for the occurrence of two-fold overlapped sub-peaks is revealed from the apparent shoulders developed mainly at the lower and higher frequency spectrum portions, at 416 nm and 478 nm, respectively. Figure 8c–h show absorption peaks for all UV-exposure durations fitted to pair Gaussian crests [18]. Mainly, Figure 8c shows the major peak at the initial *trans*-phase forming the coupled Gaussian crests. The absorption spectra exhibit high and low-frequency bands with peaks located at 416 and 474 nm with linewidths of 46.62 and 108.76 nm, respectively. Overall, the absorbance spectrum expands in its configured form as the UV-illumination exposure time changes leading to *trans*-state to *cis*-state conversion.

The amplitude and the area under the absorbance peaks are displayed in Figure 8i,j during the *trans*-phase into the *cis*-phase cycle. The area and amplitude of overall absorbance and low-frequency bands decrease continuously with time. On the other hand, the high-frequency band disappears during the first-minute exposure and is then allotted a slight decrease. The low-frequency sub-band experiences a blue-shift as a result of the light absorbed by MR. This shift’s origin comes either from an H-bond donor or a dissociated-intermediate H-bond donor root [50,51]. Therefore, the extended absorption band originates from H-bond’s mutual interaction amongst the PEO and the azo-nitrogen contents mediated by MR molecules. The shift in the amplitude of all bands amplitude certifies that the time-dependent photoisomerization process is a multi-step process [52].

Figure 9 shows the FTIR spectra for the *trans*- and *cis*-states of (PEO-BDK-MR)/I_2_ (1.0%) in the spectral range of 400–4000 cm^−1^. As was found for (PEO-BDK-MR)/I_2_ (0.1%) film, the intensive N=N band for the *cis*-MR is compared to that of *trans*-MR, indicating the enhanced photoisomerization for the *trans*-state to *cis*-state conversion.

### 4.4. Photoisomerization Kinetics of (PEO-BDK-MR)/I_2_ (10.0%)

Figure 10a shows the absorbance spectra of UV-exposed (PEO-BDK-MR)/I_2_ (10%) complex composite thin films for different exposure periods (1, 2, 3, 4, 5, 6, 7, 8, 9, and 10 min). The photoisomerization takes place via several sub-stages [17]. The two-fold bands of the absorbance spectra appear at 418 and 477 nm. For a longer exposure time (>5 min), a new peak occurs at 534 nm associated with the incorporation of I_2_ in the polymer matrix. The inset in Figure 10a shows clear steadfast hysteresis cycles with negligible loss.

The obtained photoisomerization rate and $\tau_{1/2}$ of (PEO-BDK-MR)/I_2_ (10%) complex composite thin films are found to be $p=1.10\;\times \;10^{-3}\;s^{-1}$ and 10.50 min, respectively, as demonstrated by Figure 10b. It appears that p increases and $\tau_{1/2}$ decreases as the concentration of I_2_ molecules incorporated in the polymeric matrix increases. However, $\Delta E_{a}=2.044\;\mathrm{eV}$ almost attains a constant value. The *trans*-*cis* photoisomerization adheres to the steric effect, confirming that reaction barriers are principally determined by the electronic arrangement of the −N=N− group [48]. 

Figure 10a demonstrates delayed dual-frequency bands forming two distinguished crests instead of one rough absorption peak. Figure 10c–h shows the well-known peaks of (PEO-BDK-MR)/I_2_ (10%) complex composite thin film for all UV-exposure durations in the visible range fitted to pair Gaussian crests [18]. Figure 10c shows the major peak at the initial *trans*-phase inducing the coupled Gaussian crests. The dual bands exhibit a high and a low frequency located at 418 and 477 nm with linewidths of 48.04 and 109.55 nm, respectively. The absorbance spectrum extends widely as the UV-illumination exposure time varies, as shown in Figure 10c–h, generating the transmutation from *trans*-state to *cis*-state.

Figure 10i,j show the amplitude and area under the curve of the absorbance peaks for the transmutation periods from *trans*-phase into *cis*-phase. The amplitude of the high-frequency band increases as a function of irradiation times due to the formation of $I_{3}^{-}$ inside the matrix of polymer composites. The blue-shift exhibited by the low-frequency sub-band is related to light absorbed by MR. This could be interpreted in relevance to an H-bond donor or a dissociated-intermediate H-bond donor root [50,51]. Thus, the extended absorption band occurs due to H-bond’s mutual interaction amongst the PEO and the azo-nitrogen contents throughout MR molecules. The multi-stage time-dependent photoisomerization process is indicated by the shift in the dual bands’ amplitudes [52].

Figure 11 shows the FTIR spectra for the *trans*- and *cis*-states of (PEO-BDK-MR)/I_2_ (10%) in the spectral range of 400–4000 cm^−1^. As was found for (PEO-BDK-MR)/I_2_ (0.1%) film, the intensive N=N band for the *cis*-MR is compared to that of *trans*-MR, indicating the enhanced photoisomerization for the *trans*-state to *cis*-state conversion.

### 4.5. Effect of Iodine Filler on Photoisomerization Kinetics

We examine the influence of iodine filler on the photoisomerization kinetics of photo-switchable thin films based on PEO-BDK-MR. Figure 12 shows the photoisomerization rate, $\tau_{1/2}$, and activation energy ($\Delta E_{a}$) as a function of the iodine filler concentration. The addition of iodine filler into the PEO-BDK-MR nanocomposite polymeric matrix leads to an increase in the isomerization energy barrier and increasing the processing time because iodine ion in the polymer occupies several positions randomly. Iodine takes an electron from the polymeric host atom and creates hole carriers. It can form polymer–halogen complexes when doped into polymers, and thus significantly alters their properties. When iodine is incorporated in polymers, it can occupy different sites, replace polymeric atoms in the polymer chains, or inhabit sites at the amorphous/crystalline boundaries and disperse favorably constructing charge transfer complexes that enhances electrical and dielectric properties of the host polymer. Moreover, it may exist in the form of molecular aggregates between the polymer chains. Similarly, ions aggregates might form between the polymer chains leading to a further separation of the polymeric chains further [22,54]. Thus, the neighbouring *trans*-isomers gets longer, and the intermolecular interactions reduce accordingly. Increasing the concentration of the iodine filler in the polymer composite stimulates the charge transfer between PEO-BDK-MR matrix and I_2_, iodide ($I^{-}$), tri-iodides ($I_{3}^{-}$), and molecular iodine, respectively [55,56]. The formation of iodine and triiodide ($I_{3}^{-}$) molecules inside the polymer composite materials indicates that iodide ions have been oxidized, and parts of the negative charges are transferred from more mobile smaller iodide anions onto molecular chains of PEO. The dissolution of iodide in the dielectric polymer such as PEO give rise to iodine and triiodide species [57,58,59]:$$I_{2}+\;2e^{-}\;↔I^{-}\;+{\;I}^{-}\;I^{-}+{\;I}_{2}\;↔I_{3}^{-}$$
$$I^{-}+{\;I}_{2}\;↔I_{3}^{-}$$

In this scenario, the transformation from the *trans*-to-*cis* photoisomerization is restricted to its steric effect, and consequently, the reaction barriers are fundamentally dominated by the electronic configuration of the −N=N− group [48]. The longer-time photoisomerization and higher activation energy under the iodine filler addition into the PEO-BDK-MR matrix indicate a well effective absorption of solar power with high-energy [48,60]. This fact confirms that the (PEO-BDK-MR)/I_2_ complex composite thin films may be considered potential candidates for many applications such as photochromic molecular switches, light-gated transistors, and molecular solar thermal energy storage media. 

## 5. Conclusions

In summary, fundamental mechanisms of the kinetics of photoisomerization of the (PEO-BDK-MR)/I_2_ complex composite thin films are explored, evaluated, and investigated thoroughly. Deliberately, we explore and provide new insight into the photoisomerization kinetics and time evolution for (PEO-BDK-MR)/I_2_ complex composited thin films. The kinetics of photoisomerization and time evolution of the hybrid thin film was examined via the UV-Vis and the FTIR spectroscopic techniques and by using specific analytical models. The addition of iodine filler into PEO-BDK-MR thin films leads to an increase in the isomerization energy barrier and increasing the processing time. This critical finding can be attributed to the fact that as iodine ions diffuse randomly in the polymer, they occupy several sites, such as in polymer chains, or reside at the amorphous/crystalline boundaries forming charge transfer compounds. Furthermore, Iodine may form ions aggregates between the polymer chains, increasing the distance between the two neighbouring *trans*-isomers. They become more prolonged, and the intermolecular interactions reduce accordingly. Consequently, the addition of iodine filler into PEO-BDK-MR thin films can be introduced as possible applicants for photochromic molecular switches, light-gated transistors, and molecular solar thermal energy storage media.

## Figures and Tables

**Figure 1 polymers-13-00841-f001:**
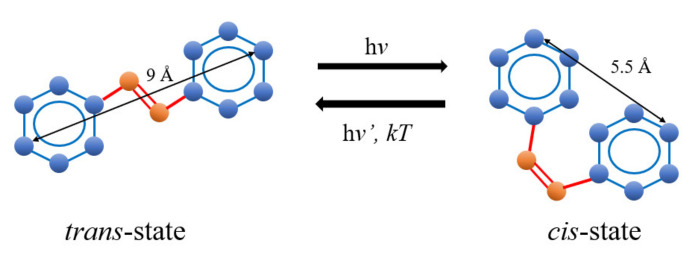
The photoisomerization process of azobenzene.

**Figure 2 polymers-13-00841-f002:**
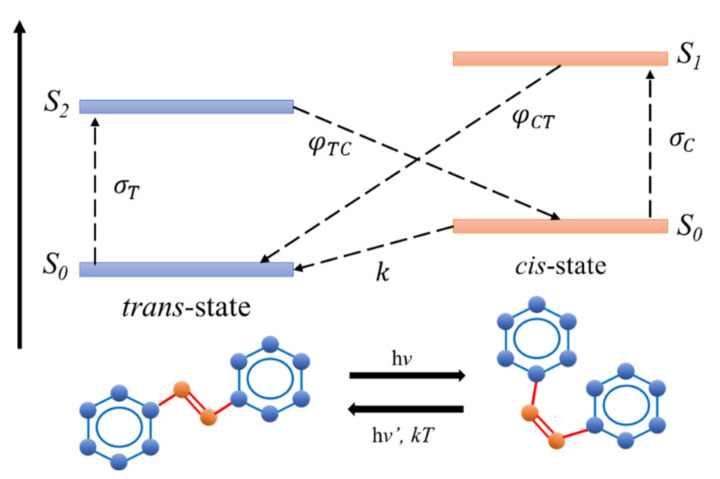
Four-level diagram illustrating the *trans*- and the *cis*-isomerization.

**Figure 3 polymers-13-00841-f003:**
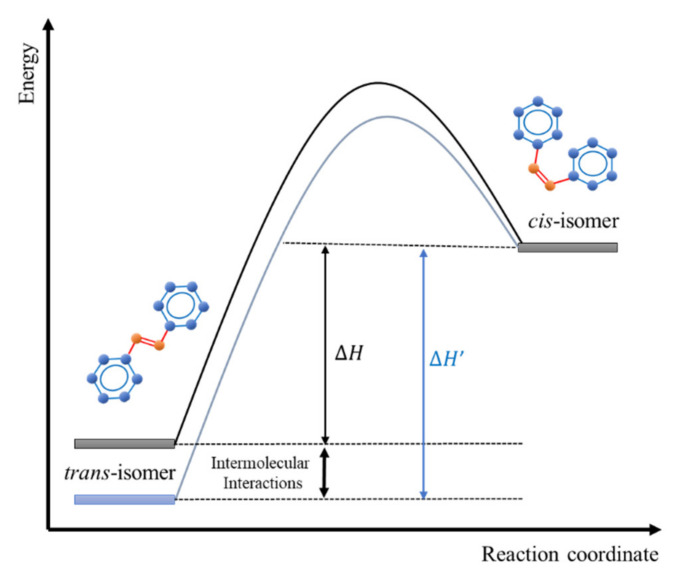
Increasing the storage energy in azobenzene molecular solar thermal storage systems by intermolecular interactions.

**Figure 4 polymers-13-00841-f004:**
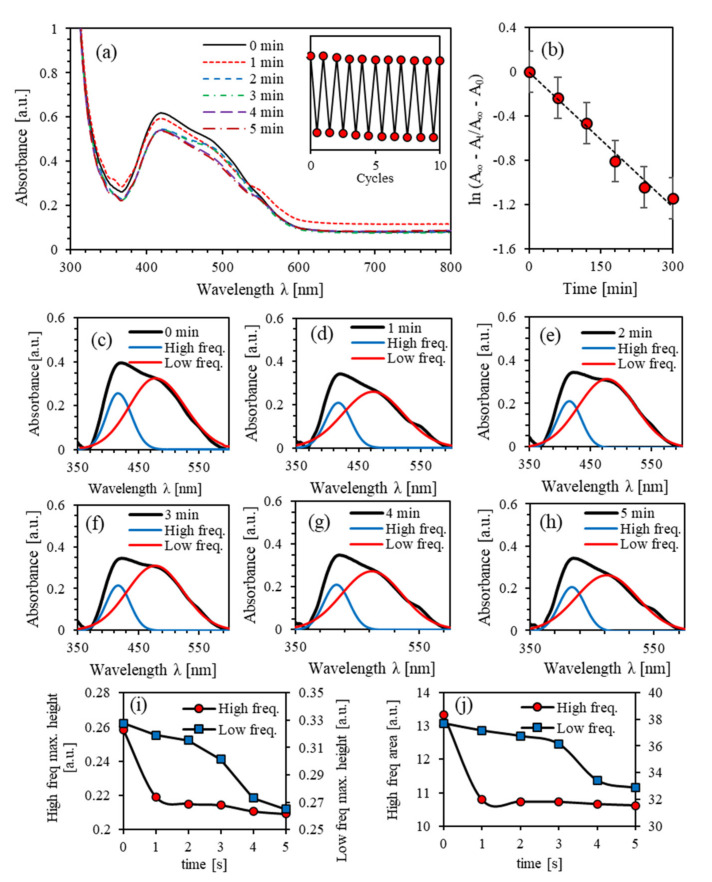
(**a**) Absorbance spectra of PEO-BDK-MR polymer composite thin film for various UV-illumination times, (**b**) the kinetic constants for *trans* → *cis* photoisomerization of PEO-BDK-MR polymer composite thin film, (**c**–**h**) Double peaks fit for the absorbance spectra of PEO-BDK-MR polymer composite thin film for various UV-illumination exposure periods of time, (**i**) The amplitude and (**j**) the integrated area-fit for the dual-peaks of the absorbance for the PEO-BDK-MR polymer composite thin film for various UV-light exposure periods of time.

**Figure 5 polymers-13-00841-f005:**
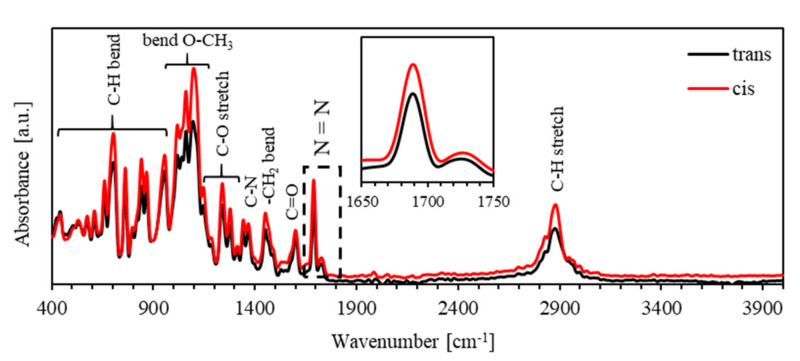
The FTIR spectra of trans- and *cis*-phases of PEO-BDK-MR polymer composite thin film.

**Figure 6 polymers-13-00841-f006:**
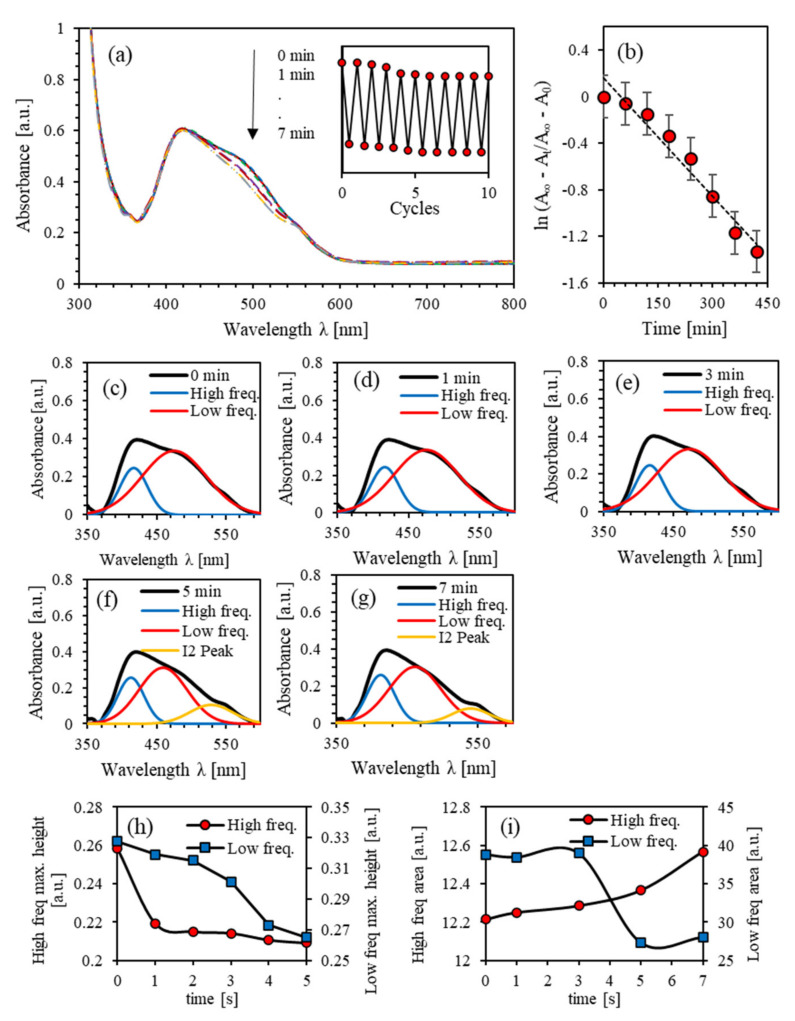
(**a**) Absorbance spectra of (PEO-BDK-MR)/I_2_ (0.1%) complex composite thin film for various UV-illumination times, (**b**) the kinetic constants for *trans* → *cis* photoisomerization of (PEO-BDK-MR)/I_2_ (0.1%) complex composite thin film, (**c**–**g**) Double peaks fit for the absorbance spectra of (PEO-BDK-MR)/I_2_ (0.1%) complex composite thin film for various UV-illumination exposure periods, (**h**) The amplitude and (**i**) the integrated area-fit for the dual-peaks of the absorbance for the (PEO-BDK-MR)/I_2_ (0.1%) complex composite thin film for various UV-light exposure periods.

**Figure 7 polymers-13-00841-f007:**
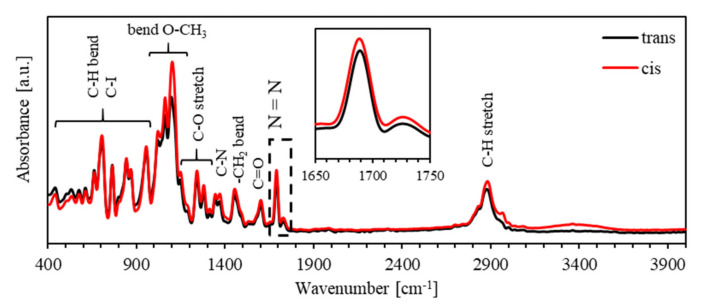
The FTIR spectra of *trans*- and *cis*-phases of (PEO-BDK-MR)/I_2_ (0.1%) complex composite thin film.

**Figure 8 polymers-13-00841-f008:**
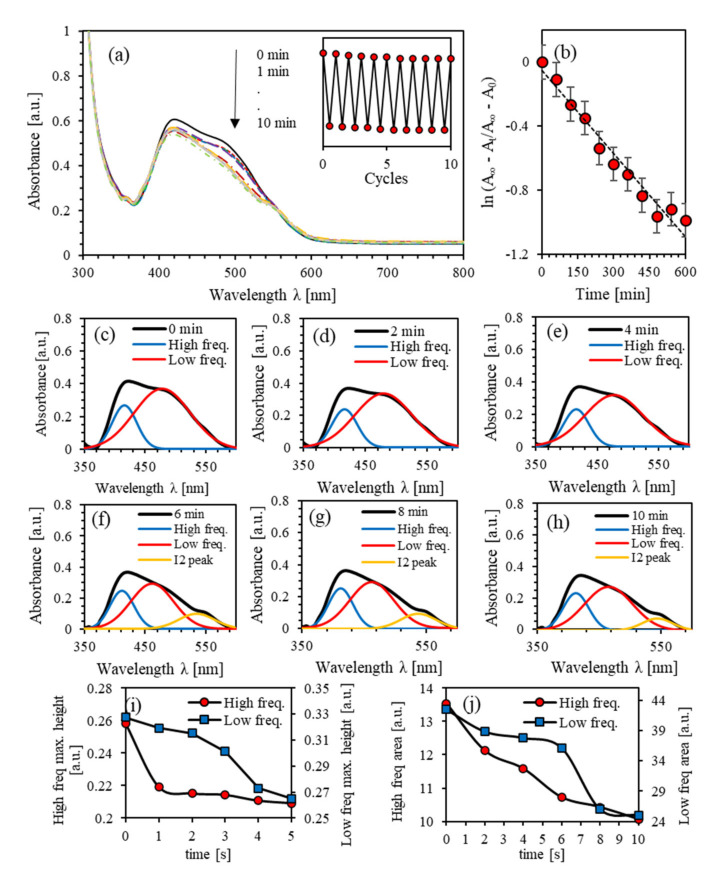
(**a**) Absorbance spectra of (PEO-BDK-MR)/I_2_ (1.0%) complex composite thin film for various UV-illumination times, (**b**) the kinetic constants for *trans* → *cis* photoisomerization of (PEO-BDK-MR)/I_2_ (1.0%) complex composite thin film, (**c**–**h**) Double peaks fit for the absorbance spectra of (PEO-BDK-MR)/I_2_ (1.0%) complex composite thin film for various UV-illumination exposure periods. (**i**) The amplitude and (**j**) integrated area-fit for the dual-peaks of the absorbance for the (PEO-BDK-MR)/I_2_ (1.0%) complex composite thin film for various UV-light exposure periods.

**Figure 9 polymers-13-00841-f009:**
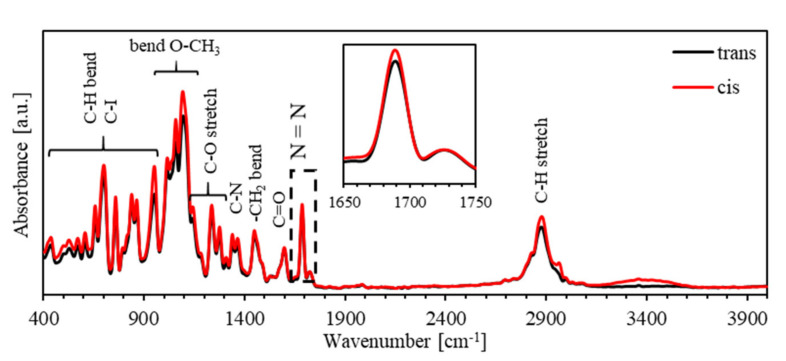
The FTIR spectra of *trans*- and *cis*-phases of (PEO-BDK-MR)/I_2_ (1.0%) complex composite thin film.

**Figure 10 polymers-13-00841-f010:**
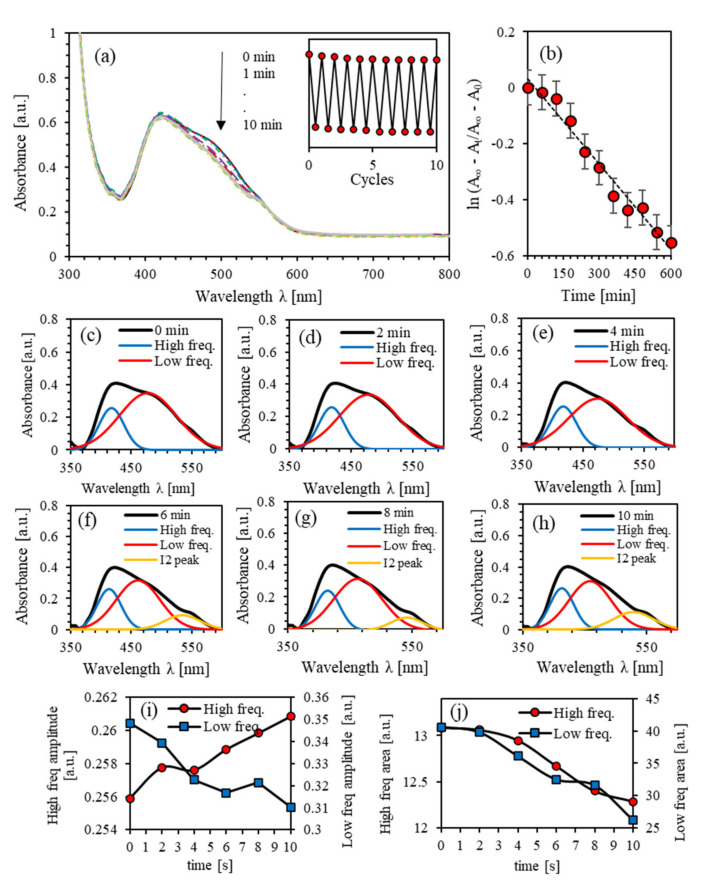
(**a**) Absorbance spectra of (PEO-BDK-MR)/I_2_ (10%) complex composite thin film for various UV-illumination times, (**b**) the kinetic constants for *trans* → *cis* photoisomerization of (PEO-BDK-MR)/I_2_ (10%) complex composite thin film, (**c**–**h**) Double peaks fit for the absorbance spectra of (PEO-BDK-MR)/I_2_ (10%) complex composite thin film for various UV-illumination exposure periods, (**i**) The amplitude and (**j**) integrated area-fit for the dual-peaks of the absorbance for the (PEO-BDK-MR)/I_2_ (10%) complex composite thin film for various UV-light exposure periods.

**Figure 11 polymers-13-00841-f011:**
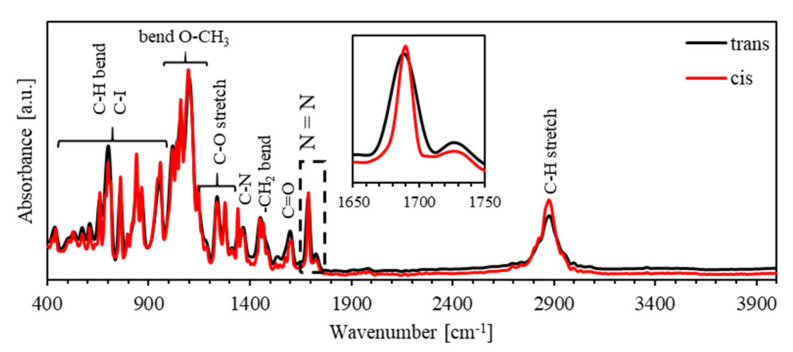
The FTIR spectra of *trans*- and *cis*-phases of (PEO-BDK-MR)/I_2_ (10%) complex composite thin film.

**Figure 12 polymers-13-00841-f012:**
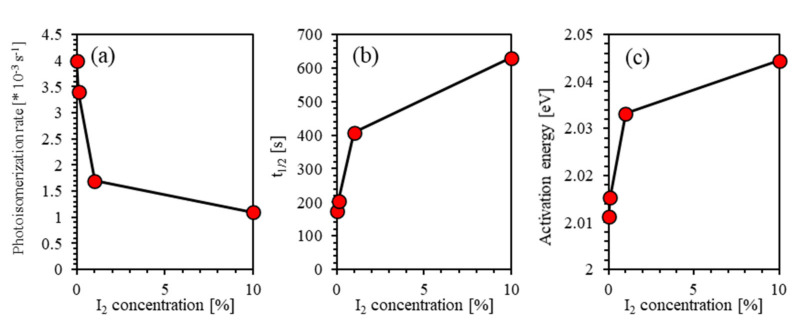
Photoisomerization rate, half-life of the *trans*-phase into *cis*-phase ($\tau_{1/2}$), and activation energy.

## Data Availability

Data availability upon request.

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
