# Peer review of "Effect of Iodine Filler on Photoisomerization Kinetics of Photo-Switchable Thin Films Based on PEO-BDK-MR"

_polymers, 2021, doi:10.3390/polym13050841_

Round 1
Reviewer 1 Report
This work investigated the role of iodine filler on the trans-cis transition for PEO-PDK-MR thin films. UV-vis and FTIR were provided to provide a quantitative and qualitative understanding of photoisomerization rate, half time, and activation energy. However, the novelty of the work is limited, and the conclusion is not well-supported by the evidence.
1. Section 4.5, Line 435-438. It is explained how iodine filler affects the isomerization process, but the following claims are not supported by the evidence:
“Examples of such positions are those in the polymer chains or the amorphous/crystalline boundaries forming charge transfer compounds.”
“ions aggregates might form between the polymer chains leading to separating the chains further.”
Line 448 – 451:
“The longer-time photoisomerization and higher activation energy under the iodine filler addition into the PEO-BDK-MR matrix indicate a well effective absorption of solar power with high-energy.”
It is suggested to provide more experimental evidence for such claims.
2. The fitting of the UV-Vis result requires further explanation. The raw data shows the absorption values at 350 nm to 600 nm are not the same. However, the authors fitting the peak by assuming the absorption at 350 nm and 600 nm are zero. It is suggested to provide more evidence and reference for the fitting protocol.
3. It is suggested that the authors revise the paper and make it more readable. The FTIR data provides many duplicated information that only needs to be mentioned once. For example, the C-H, C-I, O-CH3 peak positions do not matter that much for the main concept of the paper.
4. Line 189: “tans-state” is misspelled.
Author Response
Dear Reviewer 1,
Please kindly find the replies to your remarks and suggestions.
Please accept our regards,

Reviewer 2 Report
The manuscript has sufficiently covered the background, methods, and significance of the study, as well as the results and discussions. However, the manuscript suffers from clarity of discussion, such that, while sufficient as it is, the reader fails to understand the entire study. Figures 4 and 6, as well as 5 and 7, provide similar sets of data, that the reader could mistakenly take one as the other. The discussion accompanying these figures did not help in highlighting the results needed to be highlighted. This paper can be significantly improved by better data presentation and discussion of results, i.e. better streamlining and discussion flow.
Author Response
Dear Reviewer 2,
Please find attached the replies to your comments and suggestions.
Please accept our best regards,

Reviewer 3 Report
This paper describes photoisomerization of azobenzene moiety in iodine composite materials.
Although azobenzene is one of the most popular photochromic dyes,
formation of composite materials including it provided significance
about not only photochemistry but also redox or thin film materials.
Therefore, it is worth publishing in Polymers essentially.
There are no serious problem throughout the manuscript. However,
one point should be improved. Equation (1) in L99 and (2) in L103
were linked by the explanation of "p" parameter, though there is
not "p" in (2). So I could not follow the relationship of them.
Please improve the explanation of the equations.
That's all.
Author Response
Dear Reviewer 3,
Please find attached the replies to your comments and suggestions.
Regards,

Round 2
Reviewer 1 Report
- The author cited the reference by Palácio, Gustavo, et al. (2018) to support “the position sites in the polymers”. However, that reference discusses the coordination between ether-type oxygen atoms and ions. However, this work does not mention any coordination between iodine and the PEO phase. There is still no evidence to support the residing sites of iodine.
- The fitting protocol for UV-Vis data is not accurate here. The use of baseline correction in Origin Lab requires a specific assumption of the origin for that baseline. The method shown in reference 20, Figure 3 is more accurate.
Author Response
Dear reviewer,
We have addressed the two points you raised in the second revision of your report. Please see the attached replies.
Please accept our regards,
Prof. Ahmad Alsaad
